# Genetic Predictors of the Development of Complications after Coronary Stenting

**DOI:** 10.3390/jpm13010014

**Published:** 2022-12-22

**Authors:** Dana Taizhanova, Akerke Kalimbetova, Roza Bodaubay, Aliya Toleuova, Rakhima Toiynbekova, Zhazira Beysenbekova, Olga Visternichan, Zauresh Tauesheva, Irina Kadyrova, Dmitriy Babenko, Lyudmila Akhmaltdinova, Svetlana Kolesnichenko, Yevgeniya Kolesnikova, Olga V. Avdienko, Ainur Akilzhanova, Grigorios T. Gerotziafas

**Affiliations:** 1Department of Internal Medicine Scientific and Research Center, Karaganda Medical University, Karaganda 100000, Kazakhstan; 2Shared Resource Laboratory, Scientific and Research Center, Karaganda Medical University, Karaganda 100000, Kazakhstan; 3National Laboratory Astana, Laboratory of Genomic and Personalized Medicine, Nazarbayev University, Astana 010000, Kazakhstan; 4Cancer Biology and Therapeutics, Centre de Recherche Saint-Antoine, Institut National de la Santé et de la Recherche Médicale, INSERM U938 and Faculté de Médecine Pierre et Marie Curie (UPMC), Sorbonne Universities, 75006 Paris, France

**Keywords:** genetic predictors, risk factors, coronary stenting, complication

## Abstract

Due to the fact that there are scientific discussions about the significance of gene polymorphisms in the risk of developing cardiovascular complications after a percutaneous coronary intervention, it is of interest to evaluate the genetic predictors of the development of cardiovascular events. This study is a molecular genetic study. Association with the genes of biomarkers for inflammation and immune response increases the risk of cardiovascular events: rs1234313 (TNFSF4): (A/G, OR-4.57 (2.35–8.87), *p* ≤ 0.0001), (A/G-A/A, OR-3.14 (1.75–5.63), *p* ≤ 0.0001), and (A/G, OR = 4.01 (2.19–7.36), *p* ≤ 0.0001); rs3184504 (SH2D3); ATXN2: (C/T, OR-2.53 (1.28–5.01), T/T, OR-2.99 (1.13–7.92), *p* = 0.017)), (C/T-T/T, OR-2.61 (1.35–5.07), *p* = 0.000), and (OR-1.89 (1.15–3.09), *p* = 0.009)). According to the lipid metabolism biomarker genes, rs2943634: (A/C OR-2.57 (1.18–5.62), *p* = 0.013); according to the endothelial biomarker genes, rs2713604: (DNAJB8-AS1; GATA2): (C/T, OR-4.27 (2.35–7.76), *p* ≤ 0.0001), (C/T-C/C, OR-4.13 (2.31–7.40), *p* ≤ 0.0001), (OR-4.05 (2.24–7.30), *p* ≤ 0.0001), and (C/T, OR-3.46 (1.99–6.00), *p* ≤ 0.0001). The regression analysis found that in the presence of the rs2943634 gene polymorphism, the risk of late cardiovascular events increases by 4.007 times with 95% CI (1.502:10.692), *p* = 0.006. The genes of biomarkers for the risk of cardiovascular events are rs1234313(TNFSF4), rs3184504 (SH2D3; ATXN2), rs2943634, and rs2713604 (DNAJB8-AS1; GATA2). The only predictor of the development of new cardiovascular events was rs2943634, which belongs to the group of lipid metabolism biomarkers.

## 1. Introduction

Cardiovascular disease (CVD) continues to top the list of the top 10 causes of death. CVD has been the leading cause of death worldwide for 20 years. However, it has never taken as many lives as it does today. Since 2000, the number of deaths from CVD has increased by more than 2 million, reaching almost 9 million in 2019. Currently, CVD accounts for 16% of all deaths in the world [1].

In the Republic of Kazakhstan, diseases of the circulatory system (CVD) occupy first place in the structure of diseases among the causes of death. In 2018, 167.38 per every 100,000 people died from CVD, which was 13% lower than in 2015. Among the patients who died from CVD in hospitals, 40.4% were of working age (from 15 to 64 years). Coronary heart disease (CHD) is the leader among all CVDs, from which 11,300 people die in Kazakhstan per year (71.7 per 100,000 people), followed by acute cerebrovascular accidents (hereinafter referred to as stroke), from which 11,100 people die per year (71.8 per 100,000 people) [2].

One recent trend of interest is the study of the role of gene polymorphisms in the development of fatal cardiovascular events. Since there are scientific discussions about the significance of gene polymorphisms for the risk of developing cardiovascular complications after an intervention, it is of interest to evaluate the genetic predictors of the development of cardiovascular events.

## 2. Methods

This study is a prospective genetic validation study.

Population study. The sample of study participants was formed from among patients who applied for emergency specialized medical care in connection with Acute coronary syndrome (ACS) as well as practically healthy individuals. The selection of patients included in the study was made in accordance with the inclusion and exclusion criteria, per the protocol for the diagnosis and treatment within the Republic of Kazakhstan and international ESC recommendations. A total of 254 respondents participated in the study: 163 patients with coronary heart disease who underwent percutaneous coronary intervention, including members of both sexes over the age of 18, were included in the main group; 91 healthy controls were included in the control group.

Clinical assessments. The endpoints were new cardiovascular events: angina pectoris, recurrent MI, death from cardiovascular causes, revascularization, hospitalization for CHF, stroke, and non-coronary revascularization with indication of the timing and reasons for the end of hospitalization, arrhythmia, etc. Blood was taken from all patients for the study of molecular genetic markers.

The scientific study was approved by the ethical committee of NJSC “Medical University of Karaganda” (Protocol No. 32; dated 23 December 2019).

The material of the study was the venous blood of patients and practically healthy individuals.

Blood samples were collected in Becton Dickinson, Franklin Lakes, NJ, USA (BD Vacutainer) 2.7 mL vacutainers containing ethylenediaminetetraacetic acid (EDTA) potassium manufactured in the USA. Then, DNA was isolated from whole blood by the salting out method. The protocol for the salting out method was adopted from the authors Miller, Dukes, and Polesky, 1988 [3].

Genotyping. Genotyping was carried out by a method based on the polymerase chain reaction in real-time mode. The study method was performed in accordance with the protocol of the manufacturer. Sample preparation was performed using TaqMan^®^ OpenArray^®^ Genotyping Master Mix (Applied Biosystems, Waltham, MA, USA) for real-time PCR. Gene polymorphisms were determined using the QuantStudio TM 12K Flex Real-Time PCR system (Applied Biosystems) on TaqMan^®^ OpenArray^®^ Genotyping Plate, Custom Format 64 QuantStudio TM 12K Flex (Applied Biosystems) genotyping plates using a cardiological-profile genetic panel. The plates were filled with the reaction mixture using a QuantStudio TM 12K Flex Accufill System automated station (Applied Biosystems).

After automatic annotation and visual control, genotypes were determined for each sample (patient), which could also be edited, and the results loaded into MS Excel for further statistical data processing.

Data analysis. Statistical processing of the obtained data was carried out using the SPSS 19.0 software package.

The nature of the distribution for the normality of quantitative data was evaluated by the Kolmogorov–Smirnov criteria, since the number of observations was more than 50. Since we have 2 groups, the type of data distribution was determined for each group. With a normal distribution of quantitative data, the mean (M) and standard deviation (SD) were used for description. In case of non-normal distribution, quantitative data were described based on the median (Me) and the upper and lower quartiles (Q25, Q75).

To describe the qualitative data, the proportion of individuals with the trait of interest and the 95% confidence interval of the proportion calculated using Pearson’s χ2 method were calculated.

The Mann–Whitney U test was used to compare two independent samples on a quantitative basis. Logistic regression analysis was performed to determine the predictive value of adverse cardiovascular events.

Statistical analysis of genotyping data was carried out using the SNPStat program.

For each polymorphism included in the genotyping panel, major and minor alleles, the minor allele frequency (MAF) index, relative values for alleles and genotypes, as well as the *p* value index when calculating the Hardy–Weinberg equilibrium (HWE).

The association of genetic polymorphisms with the disease/condition was assessed according to a case–control design based on a generalized linear model (GLM), assuming underlying patterns (recessive, dominant, and log-additive).

## 3. Results

Based on the study, all quantitative variables in the main group were distributed abnormally, so we applied non-parametric criteria.

The quantitative variables in the control group (age, abdominal circumference, smoker index) were not normally distributed, but the weight variable was normally distributed.

Given that our sample in the main group had an abnormal distribution, quantitative data were described based on the median (Me) and the upper and lower quartiles (Q25, Q75). The age of half (Me) of the patients was 62 years, but a quarter of the patients was about 57 years old, while another quarter of the patients was over 68 years old. The abdominal circumference in all patients was more than 80 cm (relative to international standards), which increases their cardiovascular risk. The SBP in a quarter of patients (Q75) was more than 150 mm Hg. In addition, hypercholesterolemia (Q75) was also observed.

According to the study, Asians prevailed in both groups: 73% in the main group and 76.9% in the control group. Moreover, in both groups, there were more men than women: 68.7% and 53.8%, respectively. Respondents who were overweight prevailed in the main group, while in the control group there were more people with a normal BMI. At the same time, in the control group there were both patients who were overweight and with an obesity of 1 or 2 degrees. In accordance with the European classification of obesity in the main group, all three degrees of obesity were presented. Regarding bad habits, there were more people who smoked in the main group than in the control group. For the consumption of fats of animal origin, the maximum percentage was noted in both groups.

Based on the anamnestic data, a comparative characteristic of the general clinical data and laboratory parameters was developed, depending on gender in the main group, is presented in Table 1.

Based on Table 1, it can be argued that the respondents in the main group were ethnically heterogeneous: there were more than twice as many Kazakh men as women; while other ethnic groups did not differ significantly by gender. Taking into account the literature data on genetic polymorphisms, depending on ethnicity, the subjects were divided by race: Asians (Kazakhs); Caucasians (Russians, Ukrainians, Germans, Belarusians). More than 50% of men were overweight, while women both had a normal BMI and were overweight. In men and women in the main group, the consumption of animal fat was observed with the maximum figures: 91.1% and 90.2%, respectively.

Regarding coronary angiography, among men and women, there was a triple-vessel lesion of the coronary bed: 41.1% and 35.3%, respectively. LAD lesions were predominantly noted in both sexes, and, therefore, coronary artery stenting was performed in both men and women. Complications were noted in 55 patients (recurrence of myocardial infarction, restenosis of the coronary arteries, early post-infarction angina pectoris, life-threatening arrhythmias, total mortality, AHF, CHF, stroke), predominantly in men (33%). The most common complications in both sexes were recurrent MI, restenosis of the coronary arteries, and early postinfarction angina pectoris, which was mainly related to early cardiovascular complications. Complications were also considered according to the criteria of early complications (≥3 months) and late complications (≤3 months). Both early and late complications prevailed in men; in both cases, at a rate of more than 50%. The statistically significant differences between the sexes were age (*p* = 0.000) and smoker index (*p* = 0.005). No statistically significant differences between the sexes were found in the laboratory parameters.

Based on the literature data, we selected 53 SNPs that were associated with CAD. For analysis, we divided according to the predominant mechanisms of action: (1) the genes of biomarkers of the inflammation and immune response associated with the risk of developing cardiovascular events; (2) the genes of biomarkers of the hemostasis system associated with the risk of developing cardiovascular events; (3) the genes of biomarkers of the lipid metabolism associated with cardiovascular events; (4) the genes of endothelial biomarkers associated with cardiovascular events. The general panel of polymorphisms associated with cardiovascular events after coronary artery stenting is presented in Table 2.

As a result of statistical analysis of genotyping, it was found that association with the genes of biomarkers of inflammation and immune response increases the risk of cardiovascular events: rs1234313 of the TNFSF4 gene: codominant model (A/G, OR- 4.57 (2.35–8.87), *p* ≤ 0.0001), dominant (A/G-A/A, OR-3.14 (1.75–5.63), *p* ≤ 0.0001), and overdominant (A/G, OR = 4.01 (2.19–7.36), *p* ≤ 0.0001); rs3184504 of the SH2D3 gene, ATXN2: codominant model (C/T, OR-2.53 (1.28–5.01), T/T, OR-2.99 (1.13–7.92), *p* = 0.017)), dominant (C/T-T/T, OR-2.61 (1.35–5.07), *p* = 0.000), and log-additive (OR-1.89 (1.15–3.09), *p* = 0.009)). According to the lipid metabolism biomarker genes, rs2943634: heterozygous genotype overdominant model (A/C OR-2.57 (1.18–5.62), *p* = 0.013); rs2713604 (DNAJB8-AS1; GATA2): codominant model (C/T, OR-4.27 (2.35–7.76), *p* ≤ 0.0001), dominant (C/T-C/C, OR-4.13 (2.31–7.40), *p* ≤ 0.0001), overdominant (OR-4.05 (2.24–7.30), *p* ≤ 0.0001), and log-additive (C/T, OR-3.46 (1.99–6.00), *p* ≤ 0.0001); according to the endothelial biomarker genes, rs2713604 (DNAJB8-AS1; GATA2): codominant model (C/T, OR-4.27 (2.35–7.76), *p* ≤ 0.0001), dominant (C/T-C/C, OR-4.13 (2.31–7.40), *p* ≤ 0.0001), overdominant (OR-4.05 (2.24–7.30), *p* ≤ 0.0001), and log-additive (C/T, OR-3.46 (1.99–6.00), *p* ≤ 0.0001).

In order to determine the prognostic role of genetic polymorphism in the development of the complications and outcomes of cardiovascular events, a logistic regression analysis was performed, taking into account early and late complications. The results of the regression analysis are presented in Table 3.

Based on the regression analysis of genetic polymorphism regarding the risk of developing cardiovascular events, in the presence of rs2943634 gene polymorphism, the heterozygous AC genotype, the risk of developing late cardiovascular events increases by 4.007 times with 95% CI (1.502:10.692), *p* = 0.006. This gene is included in the group of the biomarkers of lipid metabolism disorders.

## 4. Discussion

An important role in the pathophysiology of CHD is played by the processes of inflammation and immune response. In this regard, a large number of studies have been devoted to the study of the associative relationships between the polymorphism of the genes involved in the cascade of inflammation and immune responses, the expression of these genes, and the risk of developing a cardiovascular pathology.

The TNF gene, also known as TNF-alpha, codes for a cytokine with extensive inflammatory and immune functions. Among the best-studied TNF SNPs are two found in the promoter, both of which can affect either constitutive or induced TNF expression [4].

The main metabolic changes underlying the development of CHD are lipid metabolism disorders and dyslipoproteinemia, which are controlled by various polymorphic gene variants. The predisposition to the vast majority of CHD forms is due precisely to the cumulative contribution of the many polymorphic gene variants, each of which is characterized by a relatively weak or moderate effect on lipid metabolism and the development of the disease. However, numerous studies aimed at confirming the identified associations of CHD with individual candidate genes have shown a low degree of reproducibility of results in various populations of the world. Thus, a number of researchers have also identified significant differences in the genetic structure and the contribution of individual genes to the pathogenesis of CHD in European and Asian populations [5,6]. The low reproducibility of genetic associations can be explained not only by differences in the genetic structure between populations but also by the influence of various environmental risk factors. In this regard, the study of the genetic structure of Kazakh populations in the context of its impact on the risk of developing coronary artery disease opens up wide opportunities for the large-scale screening of genetic markers and the monitoring of the main markers for the purpose of the primary prevention of the disease in the Republic of Kazakhstan.

According to the literature, the gene rs2943634 is associated with high-density lipoprotein cholesterol (HDL), and, when this polymorphism is identified in patients, the risk of ischemic stroke increases by 2.5 times [7]. According to the current study, among the patients of the main group, this polymorphism (rs2943634) is increased by 4.007 times.

## 5. Conclusions

Thus, when associated with the genes of the biomarkers of inflammation and immune response, the risk of cardiovascular events increases rs1234313 of the TNFSF4 gene: codominant model (A/G, OR-4.57 (2.35–8.87), *p* ≤ 0.0001), dominant (A/G-A/ A, OR-3.14 (1.75–5.63), *p* ≤ 0.0001), and overdominant (A/G, OR = 4.01 (2.19–7.36), *p* ≤ 0.0001); rs3184504 of the SH2D3 gene, ATXN2: codominant model (C/T, OR-2.53 (1.28–5.01), T/T, OR-2.99 (1.13–7.92), *p* = 0.017)), dominant (C/T-T/T, OR-2.61 (1.35–5.07), *p* = 0.000), and log-additive (OR-1.89 (1.15–3.09), *p* = 0.009)); according to the lipid metabolism biomarker genes, rs2943634: heterozygous genotype overdominant model (A/C OR-2.57 (1.18–5.62), *p* = 0.013); according to the endothelial biomarker genes, rs2713604 (DNAJB8-AS1; GATA2): codominant model (C/T, OR-4.27 (2.35–7.76), *p* ≤ 0.0001), dominant (C/T-C/C, OR-4.13 (2.31–7.40), *p* ≤ 0.0001), overdominant (OR-4.05 (2.24–7.30), *p* ≤ 0.0001), and log-additive (C/T, OR-3.46 (1.99–6.00), *p* ≤ 0.0001).

As predictors of the development of new cardiovascular events in the long-term period after a percutaneous coronary intervention, the prognostic criterion for the development of cardiovascular complications was the heterozygous genotype AC rs2943634 (OR-4.007 times, 95% CI (1.502:10.692), *p* = 0.006)), belonging to the group of biomarkers of the lipid metabolism disorders.

## Figures and Tables

**Table 1 jpm-13-00014-t001:** Comparative characteristics of general clinical data and laboratory parameters depending on gender in the main group.

Indicators	Male (*n* = 112)	Female (*n* = 51)	*p*
1	2	3	4
Nationality			
1—Kazakhs	84 (75%)	35 (68.6%)	-
2—Russians	17 (15.2%)	12 (23.5%)	-
3—Ukrainians	4 (3.6%)	2 (3.9%)	-
4—Others	7 (6.3%)	2 (3.9%)	-
Asians	84 (75%)	35 (68.6%)	-
Caucasians	28 (25%)	16 (31.4%)	
BMI (classification of obesity according to BMI based on European studies)			
0—Normal BMI	39 (34.8%)	19 (37.3%)	-
1—Overweight	50 (44.6%)	16 (31.4%)	-
2—Obesity of 1 degree	16 (14.3%)	7 (13.7%)	-
3—Obesity of 2 degrees	5 (4.5%)	9 (17.6%)	-
4—Obesity of 3 degrees	2 (1.8%)	0	-
Smoking			
0—No	59 (52.7%)	41 (80.4%)	-
1—Yes	53 (47.31%)	10 (19.6%)	-
Alcohol			
0—No	70 (62.5%)	43 (84.3%)	-
1—Yes	42 (37.5%)	8 (15.7%)	-
Consumption of animal fat			
0—No	10 (8.9%)	5 (9.8%)	-
1—Yes	102 (91.1%)	46 (90.2%)	-
Predisposition to coronary artery disease			
0—No	69 (61.6%)	29 (56.9%)	-
1—Yes	43 (38.4%)	22 (43.1%)	-
Stable angina			
0—None	28 (25%)	12 (23.5%)	-
FK1	46 (41.1%)	19 (37.3%)	-
FK2	29 (25.9%)	13.5 (25.5%)	-
FK3	8 (7.1%)	5 (9.8%)	-
FK4	1 (9%)	2 (3.9%)	-
Arterial hypertension			
0—None	26 (23.2%)	2 (3.9%)	-
1 degree	13 (11.6%)	7 (13.7%)	-
2 degrees	28 (25%)	13 (25.5%)	-
3 degrees	45 (40.2%)	29 (56.9%)	-
CHF			
0—None	17 (15.2%)	5 (9.8%)	-
FK1-1	60 (53.6%)	31 (60.8%)	-
FK2-2	23 (20.5%)	15 (29.4%)	-
FK3-3	12 (10.7%)	0	-
FK4-4	0	0	-
Rhythm disturbance			
0—No	92 (82.1%)	41 (80.4%)	-
1—Yes	20 (17. 9%)	10 (19.6%)	-
Diabetes			
0—No	89 (79.5%)	33 (64.7%)	-
1—Yes	23 (20.5%)	18 (35.3%)	-
Ulcer			
0—No	104 (92.9%)	48 (94.1%)	-
1—Yes	8 (7.1%)	3 (5.9%)	-
Past myocardial infarction			
0—No	29 (25.9%)	19 (37.3%)	-
1—Yes	83 (74.1%)	32 (62.7%)	-
Coronary lesion			
0—None	0	3 (5.9%)	-
1—Single-vessel	37 (33%)	15 (29.4%)	-
2–Double-vessel	29 (25.9%)	15 (29.4%)	-
3–Triple-vessel	46 (41.1%)	18 (35.3%)	-
RCA			
0—No	42 (37.5%)	19 (37.3%)	-
1—Yes	70 (62.5%)	32 (62.7%)	-
PDA			
0—No	86 (76.8%)	44 (86.3%)	-
1—Yes	26 (23.2%)	7 (13.7%)	-
LCA			
0—No	95 (84.8%)	48 (94.1%)	-
1—Yes	17 (15.2%)	3 (5.9%)	-
LAD			
0—No	26 (23.2%)	11 (21.6%)	-
1—Yes	86 (76.8%)	40 (78.4%)	-
CX			
0—No	51 (45.5%)	21 (41.2%)	-
1—Yes	61 (54.5%)	30 (58.8%)	-
OM			
0—No	77 (68.8%)	32 (62.7%)	-
1—Yes	35 (31.3%)	19 (37.3%)	-
Type of PCI (stenting)			
0—No	44 (39.3%)	25 (49%)	-
1—Yes	68 (60.7%)	26 (51%)	-
Aorta Coronary Bypass Surgery			
0—No	76 (67.9%)	30 (70.6%)	-
1—Yes	36 (32.1%)	15 (29.4%)	-
Type of PCI (balloon angioplasty)			
0—No	101 (90.2%)	44 (86.3%)	-
1—Yes	11 (9.8%)	7 (13.7%)	
Complications and outcome			-
0—No	75 (67%)	33 (64.7%)	
1—Recurrent MI, restenosis of the coronary arteries, early post-infarction angina pectoris	15 (13.4%)	(13.7%)	-
2—Life-threatening arrhythmia	(0.9%)	(2%)	-
3—Total mortality	9 (8%)	(5.9%)	-
4—AHF and CHF	9 (8%)	4 (7.8%)	-
5—Stroke	3 (2.7%)	3 (5.9%)	-
Complications	37 (33%)	18 (35.3%)	-
No complications	75 (67%)	33 (64.7)	
Early complications (≥3 months)	15 (68.2%)	7 (31.8%)	
Late complications (≤3 months)	22 (66.7%)	11(33.3%)	
Age Me (Q_25_, Q_75_)	60 (54.25;65)	66 (59;76)	0.000 *
Abdominal circumference Me (Q_25_, Q_75_)	98 (88;115)	98 (88;113)	0.837
Smoker index Me (Q_25_, Q_75_)	0 (0;20)	0	0.005 *
Syst BP Me (Q_25_, Q_75_)	130 (112.5;150)	130 (120;160)	0.074
Diast BP Me (Q_25_, Q_75_)	80 (80;90)	90 (80;90)	0.104
Cholesterol Me (Q_25_, Q_75_)	4.445 (3.3;5.5)	4.8 (3.8;5.8)	0.036
HDL Me (Q_25_, Q_75_)	1.06 (0.8;1.4)	1.0 (0.9;1.3)	0.801
LDL Me (Q_25_, Q_75_)	1.3 (1.02;2.3)	2.0 (0.98;2.7)	0.471
Triglycerides Me (Q_25_, Q_75_)	1.2 (0.9;1.8)	1.41 (1.0;2.0)	0.083
RBC Me (Q_25_, Q_75_)	4.8 (4.52;5.1)	4.41 (4.2;4.7)	
PLT Me (Q_25_, Q_75_)	230 (196;284.5)	265 (216;295)	0.081
Hb Me (Q_25_, Q_75_)	143.5 (132;153)	132 (121;138)	
WBC Me (Q_25_, Q_75_)	8.5 (6.7;11.47)	7.8 (6.5;9.0)	0.103
ESR Me (Q_25_, Q_75_)	10 (5.0;18.0)	14 (8.0;20.0)	
APTT Me (Q_25_, Q_75_)	33 (30;36.75)	32 (31;38)	0.509
Fibrinogen Me (Q_25_, Q_75_)	2.85 (2.0;3.775)	3.02 (2.3;3.8)	0.244
PTI Me (Q_25_, Q_75_)	91.5 (85;98)	95 (86;100)	0.145
PT Me (Q_25_, Q_75_)	16 (15.25;18)	16 (15;18)	0.883
FMK Me (Q_25_, Q_75_)	3 (0;3.875)	3 (0;4)	0.267
INR Me (Q_25_, Q_75_)	1.10 (1.0;1.195)	1.05 (1.0;1.16)	0.412

*—Statistically significant differences according to the non-parametric Mann–Whitney U test for comparing two independent samples on a quantitative basis.

**Table 2 jpm-13-00014-t002:** Panel of polymorphisms associated with cardiovascular events after coronary artery stenting.

№	rs	Gene Name	Abbreviation	Chromosome	Locus
1	rs1234313	*TNFSF4*	tumor necrosis factor (ligand) superfamily; member 4	1	1q25.1a
2	rs2243250	*IL4*	interleukin 4	5	5q31.1b
3	rs3850641	*TNFSF4*	tumor necrosis factor (ligand) superfamily; member 4	1	1q25.1a
4	rs4986790	*TLR4*	toll-like receptor 4	9	9q33.1c
5	rs17576	*LOC100128028; MMP9*	uncharacterized LOC100128028; matrix metallopeptidase 9	20	20q13.12b
6	rs3184504	*SH2B3; ATXN2*	SH2B adaptor protein 3; ataxin 2	12	12q24.12a
7	rs3782886	*BRAP*	BRCA1-associated protein	12	12q24.12b
8	rs1234315	*TNFSF4*	tumor necrosis factor (ligand) superfamily; member 4	1	1q25.1a
9	rs17228212	*SMAD3*	SMAD family member 3	15	15q22.33c
10	rs788016	*HSPD1*	heat shock 60kDa protein 1 (chaperonin)	2	2q33.1b
11	rs2340690	*HSPE1; HSPE1-MOB4; HSPD1*	heat shock 10kDa protein 1 (chaperonin 10)	2	2q33.1b
12	rs6725887	*ICA1L; WDR12*	islet cell autoantigen 1;69kDa-like; WD repeat domain 12	2	2q33.2a
13	rs1799963	*CKAP5; F2*	cytoskeleton-associated protein 5; coagulation factor II (thrombin)	11	11p11.2b
14	rs6025	*F5*	coagulation factor V (proaccelerin, labile factor)	1	1q24.2b
15	rs1800787	*FGB*	fibrinogen beta chain	4	4q31.3d
16	rs1799983	*CKAP5; F2*	cytoskeleton-associated protein 5; coagulation factor II (thrombin)	11	11p11.2b
17	rs2306374	*MRAS*	muscle RAS oncogene homolog	3	3q22.3c
18	rs5918	*ITGB3*	integrin; beta 3 (platelet glycoprotein IIIa; antigen CD61)	17	17q21.32a
19	rs1746048			10	10q11.21c
20	rs688034	*SEZ6L*	seizure related 6 homolog (mouse)-like	22	22q12.1a
21	rs5361	*SELE*	selectin E	1	1q24.2c
22	rs6922269	*MTHFD1L*	methylenetetrahydrofolate dehydrogenase (NADP+ dependent) 1-like	6	6q25.1b
23	rs183130	*CETP*	cholesteryl ester transfer protein; plasma	16	16q13b
24	rs1800588	*LIPC*	lipase; hepatic	15	15q21.3d
25	rs3843763	*PLTP*	phospholipid transfer protein	20	20q13.12b
26	rs268	*LPL*	lipoprotein lipase	8	8p21.3c
27	rs326	*LPL*	lipoprotein lipase	8	8p21.3c
28	rs17465637	*MIA3*	melanoma inhibitory activity family; member 3	1	1q41e
29	rs2229616	*MC4R*	melanocortin 4 receptor	18	18q21.32b
30	rs501120			10	10q11.21c
31	rs2230500	*PRKCH*	protein kinase C; eta	14	14q23.1c
32	rs2516839	*USF1; ARHGAP30; TSTD1*	upstream transcription factor 1; Rho GTPase activating protein 30	1	1q23.3a
33	rs2943634			2	2q36.3a
34	rs599839	*CELSR2; PSRC1*	cadherin; EGF LAG seven-pass G-type receptor 2	1	1p13.3b
35	rs5443	*LEPREL2; GNB3; CDCA3; USP5*	leprecan-like 2; guanine nucleotide binding protein (G protein); beta polypeptide 3	12	12p13.31d
36	rs1042714	*ADRB2*	adrenoceptor beta 2; surface	5	5q32d
37	rs8055236	*CDH13*	cadherin 13	16	16q23.3b
38	rs2774279	*USF1; ARHGAP30; TSTD1*	upstream transcription factor 1; Rho GTPase activating protein 30	1	1q23.3a
39	rs2073658	*USF1; ARHGAP30; TSTD1*	upstream transcription factor 1; Rho GTPase activating protein 30	1	1q23.3a
40	rs11206510	*PCSK9*	proprotein convertase subtilisin/kexin type 9	1	1p32.3a
41	rs383830			5	5q21.1b
42	rs5370	*EDN1*	endothelin 1	6	6p24.1b
43	rs1800779	*NOS3*	nitric oxide synthase 3 (endothelial cell)	7	7q36.1c
44	rs1800783	*NOS3*	nitric oxide synthase 3 (endothelial cell)	7	7q36.1c
45	rs1051730	*NOS3*	nitric oxide synthase 3 (endothelial cell)	7	7q36.1c
46	rs2383206	*CDKN2B-AS1*	CDKN2B antisense RNA 1	9	9p21.3c
47	rs10757278	*CDKN2B-AS1*	CDKN2B antisense RNA 1	9	9p21.3c
48	rs10116277	*CDKN2B-AS1*	CDKN2B antisense RNA 1	9	9p21.3c
49	rs1333049	*CDKN2B-AS1*	CDKN2B antisense RNA 1	9	9p21.3c
50	rs2383207	*CDKN2B-AS1*	CDKN2B antisense RNA 1	9	9p21.3c
51	rs3803	*DNAJB8-AS1; GATA2*	DNAJB8 antisense RNA 1; GATA binding protein 2	3	3q21.3c
52	rs2713604	*DNAJB8-AS1; GATA2*	DNAJB8 antisense RNA 1; GATA binding protein 2	3	3q21.3c
53	rs9536314	*KL*	klotho	13	13q13.1b

**Table 3 jpm-13-00014-t003:** Regression analysis of genetic polymorphism regarding the risk of developing cardiovascular events (the dependent variable is outcome, outcome is not).

Variable	cOR (95% CI)	*p*	aOR (95% CI)	*p*
rs2943634 (AC) Late (≥3)	0.471 (0.256—0.869)	0.016	4.007 (1.502–10.692)	0.006

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
