# Peer review of "Genetic Predictors of the Development of Complications after Coronary Stenting"

_jpm, 2022, doi:10.3390/jpm13010014_

Round 1
Reviewer 1 Report
I revised the manuscript entitled “GENETIC PREDICTORS OF THE DEVELOPMENT OF COM-2 PLICATIONS AFTER CORONARY STENTING”.
Unfortunately, I have many concerns about its possible publication.
First of all, the statement which is made in the first lines is not correct, since it is generally justified that an overall downward trend in CVD incidence and mortality rates has been made during the last decades.
The overall style of writing is not proper for a scientific paper. I mention three paradigms
line 54: “A total of 254 respondents participated in the study, of which 163 were included in the main group, patients with coronary heart disease who underwent percutaneous coronary intervention, both sexes over the age of 18, living in the Karaganda region, the control group - 91 practically healthy people”
line 57: “The endpoints were new cardiovascular events: angina pectoris, recurrent MI, death from cardiovascular causes, revascularization, hospitalization for CHF, stroke and non-coronary revasculari- zation with indication of the timing and reasons for the end of hospitalization, arrhythmia, 60 etc. d.”
line 68: “After taking the blood, the tube was inverted several times to prevent the formation of clots.”
line 224: “This study has several limitations. First, some samples were not genotyped during PCR. Second, this study was financially limited.”
In addition, there is no explanation for the abbreviations used. Surprisingly, in Table 1 Aorta Coronary Bypass Surgery is characterized as a percutaneous coronary intervention!
These are only some of the reasons I believe this manuscript do not justifies for publication.
Author Response
We would like to send the revised manuscript entitled “GENETIC PREDICTORS OF THE DEVELOPMENT OF COMPLICATIONS AFTER CORONARY STENTING".
We would like to thank the reviewers for their valuable time and helpful contributions. We appreciate the informational assistance you have provided and that their input will definitely help improve our manuscript.
We would like to respond to the comments and additions of the reviewers:
Reviewer 1
First of all, the statement which is made in the first lines is not correct, since it is generally justified that an overall downward trend in CVD incidence and mortality rates has been made during the last decades.
Our answer: Of course we agree with you. However, despite the progress and widespread use of percutaneous coronary intervention, the development of new cardiovascular events are decisive factors that limit its long-term effectiveness.
Due to the fact that there are scientific discussions about the significance of gene polymorphisms in the risk of developing cardiovascular complications after intervention, it is of interest to evaluate molecular and genetic predictors of the development of cardiovascular events.
I regret that this part was not clear in the original manuscript. We corrected the rest of the manuscript.
Thanks for the corrections and additions
We look forward to hearing from you regarding the submission of our manuscript. We will be happy to answer any additional questions and comments you may have.
Best regards,
Akerke Kalimbetova
Reviewer 2 Report
Dear Editor
I read with great interest the work by Taizhanove et al entitled "Genetic predictors of the development of complications after coronary stenting".
The search for SNPs associated with the onset, progression and complications of cardiovascular diseases is a topic that is widely addressed in the literature. To have statistical power, these studies have associated hundreds of thousands of DNAs and related clinical histories of cases and controls over time, all with the aim of minimizing: a) the false discovery rate that small studies can give and b) decreasing the "ethnic group" effect (i.e. the CARDIOGRAMplusC4D study just to cite one as example among many others).
That said, there are a few things to define:
I would organise "Methods" in sub-sections (Population study, Clinical assessments, Genotyping, data analysis etc); furthermore: - Define whether the study is prospective or retrospective - The ethics committee of which institution? - The controls are either "healthy controls" or it is not clear what "practically healthy" means - Were the SNPs selected as ? OK you used a CardioPanel ThermoFisher but how was the selection done? - Is the study aimed at building a Genetic Risk Score? If so, how would it fit into clinical practice? These are some of the critical issues to be resolved. Beyond these critical issues, the sample size remains a minor issue. If the authors claimed that this was a "validation study" on a limited population then it would make another sense.Author Response
We would like to send the revised manuscript entitled “GENETIC PREDICTORS OF THE DEVELOPMENT OF COMPLICATIONS AFTER CORONARY STENTING".
We would like to thank the reviewers for their valuable time and helpful contributions. We appreciate the informational assistance you have provided and that their input will definitely help improve our manuscript.
We would like to respond to the comments and additions of the reviewers:
Reviewer 2
I would organise "Methods" in sub-sections (Population study, Clinical assessments, Genotyping, data analysis etc); furthermore: -
Our answer: We have corrected according to your recommendation.
Define whether the study is prospective or retrospective -
Our answer: prospective
The ethics committee of which institution?
Our answer: The scientific study was approved by the ethical committee of NJSC "Medical University of Karaganda" (protocol No. 32 dated December 23, 2019).
The controls are either "healthy controls" or it is not clear what "practically healthy" means –
Our answer: On your recommendation, we changed to"healthy controls".
Were the SNPs selected as ? OK you used a CardioPanel ThermoFisher but how was the selection done? - Is the study aimed at building a Genetic Risk Score? If so, how would it fit into clinical practice?
Our answer: Based on the literature data, we selected 53 SNPs that were associated with CAD. For analysis, they were divided according to the predominant mechanisms of action. Based on the study, it is possible to identify predictors of the development of cardiovascular events in the early and late period after percutaneous coronary intervention.
We agree on the data, so the study is ongoing.
We look forward to hearing from you regarding the submission of our manuscript. We will be happy to answer any additional questions and comments you may have.
Best regards,
Akerke Kalimbetova
Round 2
Reviewer 1 Report
I revised the new version of the manuscript entitled “GENETIC PREDICTORS OF THE DEVELOPMENT OF COM-2 PLICATIONS AFTER CORONARY STENTING”.
Unfortunately, I still believe that this piece of work does not justify for publication. In the following lines, I have collected only some reasons of my concerns:
line 15: “Due to the fact that there are scientific discussions about the significance of gene polymorphisms in the risk of developing cardiovascular complications after the intervention, it is of interest to evaluate genetic predictors of the development of cardiovascular events..”. Which intervention?
line 198: “An important role in the pathophysiology of CHD and myocardial infarction is played by the processes of inflammation and the immune response.” So, CHD dos not encompass myocardial infarction too?
In addition, in table 1 there are a lot of mismatches between the columns and lines making it not comprehensive.
Author Response
Dear Reviewer
First of all, we thank you for your work and for the expertise of our article. This article is the result of a study by the team of the Medical University, that incorrect formulations were allowed in the work. Allow me to make corrections taking into account your correct comments.
1)line 15: “Due to the fact that there are scientific discussions about the significance of gene polymorphisms in the risk of developing cardiovascular complications after the intervention, it is of interest to evaluate genetic predictors of the development of cardiovascular events..”. Which intervention?
In this proposal, we are talking about percutaneous coronary intervention (PCI), mainly with the installation of a stent in the area of coronary artery occlusion.
2) line 198: “An important role in the pathophysiology of CHD and myocardial infarction is played by the processes of inflammation and the immune response.” So, CHD dos not encompass myocardial infarction too?
There is a mistake in this sentence. We are talking about coronary heart disease (CHD).
3)In addition, in table 1 there are a lot of mismatches between the columns and lines making it not comprehensive.
Inconsistencies in the table are due to technical violations. All violations have been corrected.
Thanks for your comments. The team of authors re-reviewed the correction and ask you to re-evaluate our work. Thank you again.
Best regards,
Akerke Kalimbetova
